# An Arterial Compliance Sensor for Cuffless Blood Pressure Estimation Based on Piezoelectric and Optical Signals

**DOI:** 10.3390/mi13081327

**Published:** 2022-08-16

**Authors:** Cheng-Yan Guo, Hao-Ching Chang, Kuan-Jen Wang, Tung-Li Hsieh

**Affiliations:** 1Accurate Meditech Inc., New Taipei City 241406, Taiwan; 2Department of Electronic Engineering, National Kaohsiung University of Science and Technology, No. 415, Jiangong Rd., Sanmin Dist., Kaohsiung City 807618, Taiwan

**Keywords:** Moens–Korteweg, pulse wave velocity, photoplethysmography, systolic blood pressure, diastolic blood pressure

## Abstract

Objective: Blood pressure (BP) data can influence therapeutic decisions for some patients, while non-invasive devices that continuously monitor BP can provide patients with a more comprehensive BP assessment. Therefore, this study proposes a multi-sensor-based small cuffless BP monitoring device that integrates a piezoelectric sensor array and an optical sensor, which can monitor the patient’s physiological signals from the radial artery. Method: Based on the Moens–Korteweg (MK) equation of the hemodynamic model, pulse wave velocity (PWV) can be correlated with arterial compliance and BP can be estimated. Therefore, the novel method proposed in this study involves using a piezoelectric sensor array to measure the PWV and an optical sensor to measure the photoplethysmography (PPG) intensity ratio (PIR) signal to estimate the participant’s arterial parameters. The parameters measured by multiple sensors were combined to estimate BP based on the P–β model derived from the MK equation. Result: We recruited 20 participants for the BP monitoring experiment to compare the performance of the BP estimation method with the regression model and the P–β model method with arterial compliance. We then compared the estimated BP with a reference device for validation. The results are presented as the error mean ± standard deviation (SD). Based on the regression model method, systolic blood pressure (SBP) was 0.32 ± 5.94, diastolic blood pressure (DBP) was 2.17 ± 6.22, and mean arterial pressure (MAP) was 1.55 ± 5.83. The results of the P–β model method were as follows: SBP was 0.75 ± 3.9, DBP was 1.1 ± 3.12, and MAP was 0.49 ± 2.82. Conclusion: According to the results of our proposed small cuffless BP monitoring device, both methods of estimating BP conform to ANSI/AAMI/ISO 81060-2:20181_5.2.4.1.2 criterion 1 and 2, and using arterial parameters to calibrate the MK equation model can improve BP estimate accuracy. In the future, our proposed device can provide patients with a convenient and comfortable BP monitoring solution. Since the device is small, it can be used in a public place without attracting other people’s attention, thereby effectively improving the patient’s right to privacy, and increasing their willingness to use it.

## 1. Introduction

Blood pressure (BP) is an important physiological parameter affecting patients’ treatment decisions. When the body’s BP deviates from the normal range, it may lead to many diseases, such as hypertension, which can lead to chronic kidney disease, heart disease, and other health problems [1]. For monitoring BP abnormality in the clinical setting, non-invasive ambulatory blood pressure monitoring (ABPM) represents a current useful tool. Monitoring the patient’s 24 h BP changes provides an accurate basis for the diagnosis of hypertension. It also allows patients to manage their treatment better [2]. However, ABPM requires intermittently inflating cuffs that can interrupt the patient’s daily life, resulting in discomfort and reduced treatment compliance with BP monitoring [3]. Therefore, in recent years, many researchers have developed various technologies for cuffless BP monitoring. According to the definition of a cuffless BP monitor in the Institute of Electrical and Electronics Engineers (IEEE) standard, a cuffless BP monitor is a BP monitor without an occluding cuff [4]. This implies that the difference between a cuffless BP monitor and a traditional BP monitor is that there is no need to pressurize the patient’s arteries, thereby reducing the discomfort caused by measurement. In the early days, popular cuffless BP monitoring technologies used electrocardiography (ECG) and PPG to measure the pulse arrival time (PAT) by calculating the time difference between the two [5,6]. Since PAT is correlated with arterial compliance in the description of hemodynamics, BP can be estimated under calibration by regression methods [7]. However, in recent studies, researchers found that the regression method cannot accurately estimate BP due to the influence of the pre-ejection period (PEP) on PAT [8]. Pulse transit time (PTT) is the time difference between pulse waves at two positions on the local artery. The speed of pulse wave transmission is faster than the speed of blood movement. Notably, measuring the PTT of an arterial segment can reflect the correlation of intra-arterial pressure [9]. However, PTT does not have the problem of PAT being affected by PEP and, thus, can avoid the problem of PAT being unable to accurately estimate BP. At present, related studies are using PTT to estimate BP. Piezoelectric sensors are used to measure the time difference between the wrist and fingers due to the pulse wave generated by the cardiac cycle [10]. In this method, an optical sensor array is placed on the finger to measure the change in arterial blood volume and calculate the time difference after taking the derivative of the alternating current (AC) signal generated [11]. The BP sensing patch integrates a flexible piezoresistive sensor (FPS) and epidermal electrocardiogram (ECG) sensor [12]. A sensor measures the carotid pulse wave by using the tissue movement generated by the pulse wave to change the magnetic field [13]. A probe integrates two ultrasonic transducers to measure the time difference between the echo signals of local arterial changes [14]. The method of measuring the time difference of the bioimpedance of the blood volume change in the artery is used [15]. Moreover, other sensors have the potential for cuffless BP measurement [16]. Flexible sensors can function through a chemical reaction and physical sense, such as biodegradable ferroelectric skin sensors [17], triboelectric self-powered sensors [18], and microfluidic sensors [19]. Many related studies have shown that various technical means of measuring PTT have been developed. However, in these studies, estimating BP from PTT still involved using the regression method to calibrate the BP curve (i.e., directly converting PTT to generalized PWV and modeling a regression model with PWV-BP) [20]. In recent years, due to breakthroughs in machine learning, some studies of cuffless methods have also tried to use machine learning methods to estimate BP [21]. However, machine learning models need different hyperparameters to predict blood pressure, including demographics (e.g., age and gender). Nevertheless, researchers are unclear how much of the attained BP measurement accuracy, especially of the calibration-free device, is due to the actual hemodynamic measurement [22]. Additionally, the computing loading of deep learning is unsuitable for embedded systems with low computing power.

The calibration curve relating PTT to BP depends on blood density, arterial stiffness, and geometry. Except for blood density, other arterial parameters are patient-specific, with experimental data suggesting that the calibration curves vary among individuals [23]. Therefore, a calibration curve customized to each patient will be optimal. However, the disadvantage of this method is obvious, since the operational complexity of the calibration procedure will increase. In the theory of hemodynamics, the potential variables affecting BP have other contributions from arterial parameters in addition to PWV; thus, the regression model of BP cannot be applied to all patients. There are two reasons why the direct use of PTT and BP to model a regression model for estimating a patient’s BP is inaccurate. The first is that arterial parameters affect BP and differences in the arterial characteristics of different patients may cause regression models to fail. The second is that the distance for measuring the PTT is short, and the error generated by the conversion to the PWV difference will increase [24]. Since the regression model is sensitive to PTT error, the result of BP estimation can easily produce a larger distribution range. This study proposes a system that uses a combination of piezoelectric sensor arrays for PTT measurement and near-infrared spectroscopy (NIRS) to measure a patient’s photoplethysmography (PPG) intensity ratio (PIR) by using NIRS to correlate the PIR with the patient’s arterial parameters. We used arterial parameters to calibrate the PTT and BP models based on hemodynamics, which considers the neglected components in previous relevant studies and reduces the error of BP estimation. Finally, we used the ISO 81060-2 standard to evaluate the performance of the hemodynamic method compared to regression models.

### Our Contribution

This study developed a small cuffless BP measurement device to verify cuffless BP monitoring (see Figure 1). This device includes the following two parts: a motherboard and a sensor module. The sensor module is connected to the motherboard through a flexible printed circuit (FPC). The motherboard includes a microcontroller unit (a STM32F405, produced by STMicroelectronics Geneva, Switzerland), a wireless communication chip (a NRF52832, produced by Nordic Semiconductor Inc. Trondheim, Norway), and an analog front end (an AD8669, produced by Analog Devices Inc., Wilmington, MA, USA). The sensor module consists of two piezoelectric sensors and a NIRS optical sensor (a MAX30101, produced by Maxim Integrated Inc., San Jose, CA, USA). The analog signal of the piezoelectric sensor is input to the motherboard’s analog front end (AFE), while the optical sensor is connected to the inter-integrated circuit (I2C) interface of the microcontroller unit (MCU). When the patient has been measured (Figure 2), the cuffless BP monitor (Figure 2a) will transmit the results to a smartphone, which displays the patient’s BP data and records it (see Figure 2b).

We used this cuffless BP monitor to measure 20 adult participants according to the ISO 81060-2 protocol. We estimated BP using the following two methods: a regression model using PTT with BP, and arterial parameters combined with PTT to estimate BP using the P–β model (derived from the MK equation). In the results of this paper, we compare the mean difference and standard deviation of the two methods for the gold standard, assess whether the performance of the P–β model for cuffless BP measurement can be improved, and outline relevant discussions and future work directions based on the results.

## 2. Materials and Methods

Figure 3 presents the cuffless BP monitor proposed in this study, which uses a multi-sensing module that integrates piezoelectric and optical sensors. Figure 3a–c show the sensor module, which integrates two piezoelectric sensors and one PPG sensor, with the PPG sensor being placed between the two piezoelectric sensors. This is because when two piezoelectric sensors sense the pulse wave of the radial artery, it can ensure that the PPG sensor is placed above the radial artery. If the pulse wave amplitude measured by one of the two piezoelectric sensors does not exceed the threshold, the sensor module may not be placed parallel to the radial artery. Figure 3d shows that the sensor module must be placed parallel above the radial artery. After the two piezoelectric sensors measure the pulse wave in the same cardiac cycle, the pulse wave at two different positions in the radial artery segment can be calculated as the time difference to obtain PTT. Optical sensors can measure optical tissue information, such as changes in blood volume in the radial artery, to estimate arterial parameters.

### 2.1. Pulse Wave Detection Sensor

Figure 4 is a schematic diagram of the structure of the piezoelectric sensor and the pulse wave transmitted by the radial artery. We used the combination of an O-ring and the assembly of mechanical structures to perform an interference fit on the piezoelectric ceramics so that the inside of the sensor chamber is airtight. Additionally, a hole was opened at the top of the mechanical structure. The signal line of the piezoelectric ceramics was connected to the AFE of the motherboard through this hole. Part of the opening hole was also sealed with airtight glue. The rubber membrane in contact with the skin surface was also assembled with the chamber for an interference fit so that the pressure and volume within the sensor cavity remain stable. When a pulse wave is transmitted to the sensor, the rubber membrane will be moved by the pressure *P_a_* due to the movement of the radial artery and the surrounding tissue so that the internal pressure *P_A_* of the cavity increases and changes, resulting in a pressure difference Δ*P*, as in Equation (1).
(1)$$\Delta P=P_{A}+P_{a}$$

The volume *V* within the chamber is the move distance *L* into the sensor generated by the rubber membrane and the cross-sectional area *A* of the rubber membrane itself. The volume value calculated by the product of *L* and *A* is the volume [25] inside the compression chamber, which makes the original volume change Δ*V*, as in the following Equation (2):(2)ΔV=V−A·L

Here, Δ*V* and *P_a_* can be expressed as the relationship of pressure resulting from a change in volume according to Equation (3), where the resulting equation involves a constant *γ*, the ratio of specific heats, which is approximately 1.4 for air [26].
(3)$$\frac{P_{a}}{P_{A}}=-\gamma \frac{\Delta V}{V}=-\gamma \frac{A\cdot L}{V}$$

Since the mass m of the volume compressed by the rubber membrane into the chamber is A·L multiplied by the air density ρ, as in Equation (4), through Newton’s law of acceleration, as in Equation (5), the force *F* acting on the piezoelectric ceramics and the difference of the acting force can be known. This relationship is expressed in Equation (6).
(4)m=ALρ
(5)$$\frac{F}{m}=\frac{d^{2}x}{dt^{2}}$$
(6)$$\frac{d^{2}x}{dt^{2}}=\frac{P_{a}S}{SL\rho }=-\frac{\gamma SP_{A}}{\rho VL}x$$

Piezoelectric ceramics are deformed by pressure. The charge is generated in the piezoelectric ceramics according to the direct piezoelectric effect. We can detect the pulse wave signal through the change in the charge [27]. Piezoelectric ceramics detect that the signal generated by the pulse wave is usually in the order of millivolts. The AFE on the motherboard can pre-process the pulse wave signal to improve the signal-to-noise ratio (SNR) of the pulse wave signal. Figure 5 presents the topology of the AFE on the motherboard, while Figure 5a presents the voltage follower (VF). Since the piezoelectric ceramics’ parasitic capacitance *C_p_* is 8000 pF (at 120 Hz) [28], *R*1 is external. When the load resistor forms a high-pass filter, the relationship between the cut-off frequency ƒ and the external load resistor is calculated using Equation (7). If *R*1 uses a 20 MΩ resistor, the cut-off frequency is approximately 0.1 Hz. When *R*2 is equal to *R*1, a direct current (DC) offset circuit [29] is formed to increase the voltage level of the pulse wave signal. At this time, the non-inverting input voltage of VF is represented by Equation (8), where *V_p_* is the pulse wave detected by the piezoelectric ceramics’ generated voltage. Figure 5b presents a signal amplifier that can amplify the pulse wave signal to improve the SNR. It is possible to use *C*1 to adjust the gain of the amplifier [30]. In this study, an amplification of approximately 18 times is used for gain, calculated as represented in Equation (9), to determine the gain range of the pulse wave signal. Figure 5c is a second-order Sallen–Key low-pass filter (LPF) [31]. Equation (10) is its transform function calculation formula, the cut-off frequency *ƒ*_*c*(*LPF*)_ is determined by Equation (11), and the cut-off frequency of LPF is 10.6 Hz. Here, *R*5 and *R6* are the same resistance value. Additionally, *C*2 and *C*3 also have the same capacitance value, which can attenuate the high-frequency noise amplified by the amplifier. After the AFE pre-processes the signal, it is sampled and converted into a digital signal by the motherboard’s analog-to-digital convert (ADC), as shown in Figure 5d. In our previous study, we developed a series of digital signal processing algorithms to process the piezoelectrically sensed pulse wave signal, which can effectively detect the pulse wave in normal heart rhythm and arrhythmia patients [32]. This study uses the same method to process the pulse wave and calculate the PTT.
(7)$$f=\frac{1}{\left(2\cdot \pi \cdot R1\cdot C_{p}\right)}$$
(8)$$V_{in}=V_{p}+\frac{VCC\cdot R1}{\left(R1+R2\right)}$$
(9)$$V_{out}=V_{in}\cdot (\frac{R4}{R3+\frac{1}{\omega \cdot C1}}+1)$$
(10)$$\frac{V_{out}(s)}{V_{in}(s)}=\frac{\frac{1}{R5\cdot C2\cdot R6\cdot C3}}{s^{2}+s\left(\frac{1}{R6\cdot C2}+\frac{1}{R5\cdot C2}\right)+\frac{1}{R5\cdot C2\cdot R6\cdot C3}}$$
(11)$$f_{c}{}_{\left(LPF\right)}=\frac{1}{2\pi \sqrt{R5\cdot C2\cdot R6\cdot C3}}$$

The 12-bit ADC of the motherboard samples the pulse wave signal of the AFE at a sampling rate of 5 kHz and uses the finite impulse response (FIR) to filter the signal [33]. The FIR filter algorithm we use is a band pass filter (BPF), where the cut-off frequency for high-pass filtering is 0.7 Hz and the cut-off frequency for low-pass filtering is 9.5 Hz [34]. After the pulse wave signal is pre-processed by digital filtering, the dynamic threshold algorithm can effectively detect the characteristic points of the pulse wave, as shown in Figure 6. Finally, the PTT (inversely proportional to PWV) can be obtained by calculating the pulse wave time difference between two positions of arterial segments [35].

### 2.2. PPG Signal Process and Arterial Parameter Estimates

To improve the accuracy of the BP estimation model in this study, it was necessary to measure the patient’s arterial compliance. We used the PPG sensor to quantify the patient’s arterial dimensions and arterial change. The PPG sensor we used is a diffuse reflectance spectroscopy (DRS) sensor that integrates three wavelengths of light-emitting diode (LED) and a photodiode (PD) [36] to ensure that the PPG sensor is placed on the skin surface parallel to the radial artery. When the piezoelectric sensor array of the cuffless BP monitor detects the pulse wave, it will determine that the SNR of the pulse wave qualified the set threshold, indicating that the device is placed on the artery. At this point, the PPG sensor is activated to sense the optical signal of arterial change. When the photons excited by the PPG sensor enter from the epidermal tissue and return to the PD, the optical information of the tissue can be obtained [37]. The depth of the photons entering the tissue depends on the distance of LED to PD and wavelength. To measure the optical characteristics of the tissue as deeply as possible, we chose infrared light with a wavelength of 880 nm as the PPG signal for our estimation of arterial changes. The ADC sampling rate is 100 Hz and the resolution is 18 bits, because infrared light can enter the tissue more than visible light. The deeper the depth, the more tissue optical information can be obtained [38,39]. The distance between the LED and PD is 3 mm. Since the light path of the photon in human tissue is banana-shaped [40], the detection depth can be greater than 3 mm, which affects the radial artery depth. The main factor is the body mass index (BMI) [41]. In most cases, the radial artery depth is approximately 2.5 mm in individuals with a BMI of less than 30 [42,43]. After determining that the PPG sensor has known physical parameters and the radial artery’s physiological characteristics, we used an ultrasonic imaging device as a reference instrument to model the arterial parameters measured by ultrasonic imaging and the quantized PPG signal [44]. The instruments we used included an ultrasonic imaging device (a uSmart 3300, produced by Terason, Burlington, MA, USA) and an L-type probe (a 16HL7, produced by Terason Inc., Burlington, MA, USA) to sample the radial artery parameters of the participants. The researcher operated the L-type ultrasound probe to collect an ultrasound image of the participant’s radial artery and measured the depth, diameter, change, wall thickness, and other information (see Figure 7).

In near-infrared (NIR) tissue spectroscopy, the modified Beer–Lambert law is usually used to detect the mean path lengths of photons in tissue [45]. The optical information included by the tissue can be obtained through the transmission, absorption, and reflection of photons after moving in the tissue. The measurement technology of blood oxygen involves creating a regression model of the blood oxygen curve through the absorption ratio of hemoglobin and oxygenated hemoglobin for different wavelengths [46]. Monte Carlo (MC) simulations and animal experiments (using fresh porcine carotid arteries) have shown that changes in the PPG intensity relative to its mean value can be used to estimate the size of the absorbing medium [47]. The larger the arteries, the higher the absorption rate of the DC component of the PPG signal; thus, the lower the DC value. Additionally, the blood volume changes greatly when the pulse beats, and the amplitude of the AC component of the PPG signal increases. According to this principle, we can use Equation (12) to create a regression model to estimate arterial diameter by using the normalized PPG signal and the arterial dimensions of the ultrasound image. Since the change in end-diastolic diameter *D_d_* is relative to the instantaneous diameter *D* and differs by less than 10% during a cardiac cycle [48], we use *D_d_* to model the estimation of arterial dimensions. The volume change in the artery due to the pulse will make the intensity of the PPG signal have the highest level *I_H_* and minimal level *I_L_* changes (see Figure 8). The modified Beer–Lambert law can express the change in arterial dimensions and the relationship between *I_H_* and *I_L_* in Equations (13) and (14) [49] as *α_DC_* = ε·c, where ε is the absorbance coefficient of human tissue and c is the tissue and the material concentration of blood. Here, *d_DC_* is the DC component of the light path of photons in human tissue, including the optical properties of the scattering of each tissue layer due to different media. The product of the two contributes to the DC component of PPG. Additionally, *I*_0_ is the initial light intensity. Since *D_s_* and *D_d_* represent changes in arteries with the cardiac cycle, through Equation (15), *I_H_* and *I_L_* can be expressed as a function of PPG signal to arterial change, while *I_H_* and *I_L_* can be expressed as a ratio (i.e., PIR). Therefore, we can express the systolic diameter *D_s_* and end-diastolic diameter *D_d_* of the artery using the formula of Equation (16) [50]. Thus, the normalized PIR signal and the characteristic parameter α create a regression model with the arterial change volume. Using this model, the subject’s arterial change volume Δ*D* can be estimated. Notably, this study used the PPG signal and arterial compliance for regression modeling, with arterial parameters being measured between 2 and 6 cm above the styloid process of the patient’s wrist [51].
(12)$$PPG_{norm}=\frac{\ln(AC_{\lambda })}{\ln(DC_{\lambda })}$$
(13)$$I_{L}=I_{0}\cdot e^{-\alpha_{DC}\cdot d_{DC}}\cdot e^{-\alpha \cdot D_{s}}$$
(14)$$I_{H}=I_{0}\cdot e^{-\alpha_{DC}\cdot d_{DC}}\cdot e^{-\alpha \cdot D_{d}}$$
(15)$$\Delta D=D_{s}-D_{d}=\frac{\ln(\frac{I_{H}}{I_{L}})}{\alpha }$$
(16)$$PIR=\frac{I_{H}}{I_{L}}=e^{-\alpha \cdot \pi \cdot D\cdot (D_{s}-D_{d})}$$

### 2.3. Cuffless BP Model

According to the principle of hemodynamics, *PTT* can be obtained by measuring the pulse wave time difference between two points on the local artery, which can be expressed as Equation (17), where *L* is the distance of the sensor’s position on the artery.
(17)$$PWV=\frac{L}{PTT}$$

According to the Bramwell–Hill (BH) model [52], the pressure pulse in the artery will generate a local *PWV*, as in the following Equation (18):(18)$$PWV=\sqrt{\frac{V\cdot dP}{\rho \cdot dV}}$$

The modified BH model can be expressed as a function of the pulse pressure (PP) Δ*P* relative to the change in the arterial cross-sectional area. Here, *V* is the initial volume, Δ*V* and Δ*P* are the volume changes in the arterial segment for PP, and *ρ* is the blood density, typically using a whole blood density of 1.06 g/mL [53]. The BH model is derived from the MK equation [54]. The MK equations of Equations (19) and (20) describe the relationship between *PWV* and BP, where *D* is the diameter, *h* is the wall thickness or intima-media thickness (IMT), ρ is the whole blood density, *E* is Young’s modulus, *P* is pressure, and *E*_0_ and *γ* are coefficients of Young’s modulus [55].
(19)$$PWV=\sqrt{\frac{E\cdot h}{D\cdot \rho }}$$
(20)$$E=E_{0}\cdot \exp^{\xi \cdot P}$$

Based on Equation (21), we know that the pressure curve of Young’s modulus and PWV are related to arterial compliance. We mathematically rewrite Equations (19) and (20) to obtain Equation (22), which can theoretically be directly calculated by the MK equation for B. Notably, *P* approximates mean arterial pressure (MAP) [56]. Here, *ξ* is a constant that depends on the particular vessel.
(21)$$E=\frac{\rho \cdot D\cdot PWV^{2}}{h}$$
(22)$$P=\frac{\ln(\frac{PWV^{2}\cdot \rho \cdot D}{h\cdot E_{0}})}{\xi }$$

Since arterial parameters are not usually easy to measure, recent popular technical studies related to cuffless BP monitors have been based on the MK equation, which creates a regression model for the correlation between *PTT* and blood pressure [57,58,59]. This type of regression can take many changes in form (e.g., the parameters can be linear or non-linear). This paper uses a popular regression model, namely Equation (23) for performance comparison. Here, *K*_1_, and *K*_2_ is the regression coefficient.
(23)$$P=K_{1}\cdot \ln(PTT)+K_{2}$$

To evaluate whether introducing arterial compliance into the cuffless BP estimation model can be more accurate than the regression-only approach, we had to use the P–β model based on hemodynamics [60]. The P–β model is derived from the MK equation and the BH model. This model relates *PWV* to Δ*P*. As in Equation (24), we can estimate the Δ*P* of the local artery [61] using the *PWV*, Δ*D*, *D_d_*, and ρ (using 1.06 g/mL) (PP).
(24)$$\Delta P=\rho \cdot PWV^{2}\cdot (2\frac{\Delta D}{D_{d}}+{\left(\frac{\Delta D}{D_{d}}\right)}^{2})$$

The intra-arterial MAP can be estimated by Equation (25), where *β* is a constant that reflects the local arterial stiffness during a cardiac cycle, independent of changes in arterial pressure during the measurement cycle [62,63]. After obtaining *MAP* and PP through the P–β model, we can track the systolic BP (*SBP*) and diastolic BP (*DBP*) through the calculation method of Equation (26), where the coefficient *k* needs to be adjusted according to the results of the experiments. In this study, 0.76 was the value for *k* [64].
(25)$$P=\frac{2\cdot \rho \cdot PWV^{2}\cdot D_{d}}{\beta D}$$
(26)$$\{\begin{array}{c} DBP=MAP\cdot k\\ SBP=MAP+(1-k)\cdot DBP \end{array}$$

## 3. Results and Discussion

### 3.1. Experiment Result

We experimented with 20 adult participants using the cuffless BP monitor proposed in this study. The cuffless BP monitor measured participants’ PTT and arterial parameters for 25 s by using regression and P–β models for BP estimation, respectively. This comparative study followed ANSI/AAMI/ISO 81060-2:2018 [65]. We used the Welch Allyn 5098-27 DS66 trigger aneroid sphygmomanometer for reference BP measurement.

Figure 9 presents the two cuffless BP estimation models applied to the distribution of BP statistics of the same participants. The regression model has a more significant measurement dispersion, and the maximum value is also significantly higher than that of the P–β model.

Based on the comparison of the Bland–Altman analysis of SBP (Figure 10), the results of the regression model method indicated a SBP of 0.32 ± 5.94. The results of the P–β model method (Figure 11) indicated an SBP of 0.75 ± 3.9.

Based on the comparison of the Bland–Altman analysis of MAP (Figure 12), the results of the regression model method indicated a MAP of 1.55 ± 5.83. The results of the P–β model method (Figure 13) indicated a MAP of 0.49 ± 2.82.

Based on the comparison of the Bland–Altman analysis of DBP (Figure 14), the results of the regression model method indicate a DBP of 2.17 ± 6.22. The results of the P–β model method (Figure 15) indicate a DBP of 1.1 ± 3.12. According to the results, it was found that the dispersion of BP values estimated by the regression model is greater than that of the P–β model. The SBP value deviation is slightly greater than ±2 SD, while those of the MAP and DBP values are slightly greater than ±3 SD. Additionally, the error mean of the BP estimation for the P–β model is smaller than that of the regression model. As in Table 1, the cuffless blood pressure monitor which can pass the measurement performance requirements of criterion 1 and 2 of ANSI/AAMI/ISO 81060-2:2018. Moreover, using the P–β model is better than using the current popular regression model methods since it is more concentrated and accurate. According to the Pearson Correlation analysis, as shown in Figure 16, Figure 17 and Figure 18, the r value of SBP and MAP is 0.98, and the DBP is 0.97, indicating that the data sets are equal or very close to it.

### 3.2. Limitations and Future Works

Although a method of measuring arterial parameters using piezoelectric sensors combined with optical sensors is proposed, we also compared the current popular cuffless BP measurement regression model and the P–β model. Our results indicate that the P–β model introduces arterial compliance for calculation, while the accuracy of BP estimation is greater than that of the regression model. However, there are still some limitations that require further research and improvement. The piezoelectric sensor measures the PTT of the local arteries, and the error is amplified when converted to PWV due to the short distance, which may lead to errors. Therefore, in practice, we must take the median of the PTT in the measurement period as the representative parameter of the BP estimation. In the future, to achieve beat-to-beat BP measurement, the design of the mechanical structure must be improved to reduce the measurement error, which will make the beat-to-beat BP measurement possible. To be able to use the P–β model to estimate BP, we used the PIR technology of the PPG sensor to quantify the arterial parameter. However, at present, we must use an ultrasound imaging device to determine the patient’s arterial depth to ensure the depth range that the PPG sensor can measure in the patient’s artery. Moreover, future research should include more patients with different tissue characteristics, such as those with a BMI greater than 40 and those who have more adipose tissue (resulting in greater arterial depth).

## 4. Conclusions

This study involved designing and implementing a cuffless BP monitor combining piezoelectric and optical sensors. Here, we compared the performance of two BP estimation models, namely the popular regression model and the P–β model. According to the results, we verify that the P–β model introduces arterial compliance in BP estimate and is more accurate than the regression model in terms of the accuracy of BP estimation. Notably, it also passes the ANSI/AAMI/ISO 81060-2:2018 performance indicators. In addition to its small size and low cost, our proposed cuffless BP monitor also uses more physiological parameters for BP estimation, thereby improving accuracy. Compared to most studies of cuffless technology, our system design can be integrated into wearable devices or applied in certain BP monitoring scenarios involving small spaces. Additionally, in comparison to traditional BP monitoring approaches, our design is easier and more comfortable to use, which can result in improved patient BP monitoring compliance while making the diagnosis of hypertension more reliable.

## Figures and Tables

**Figure 1 micromachines-13-01327-f001:**
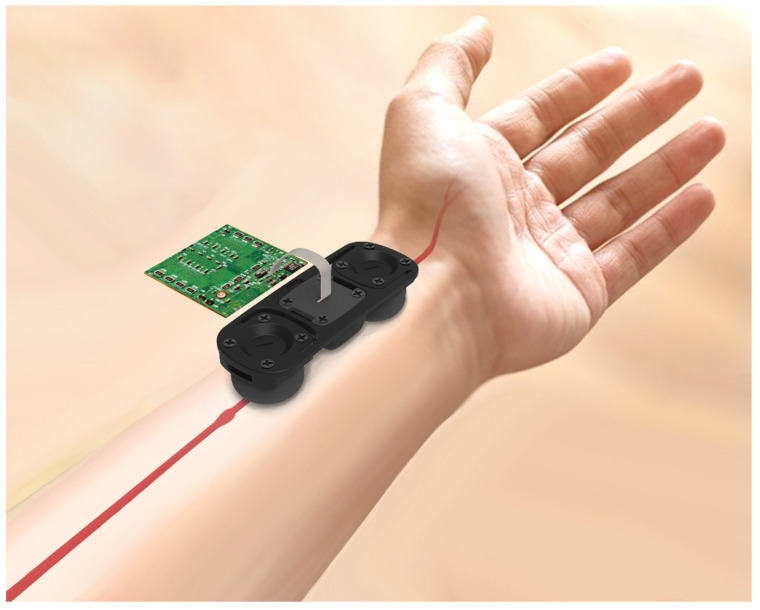
A small cuffless BP measurement device used to verify cuffless BP monitoring.

**Figure 2 micromachines-13-01327-f002:**
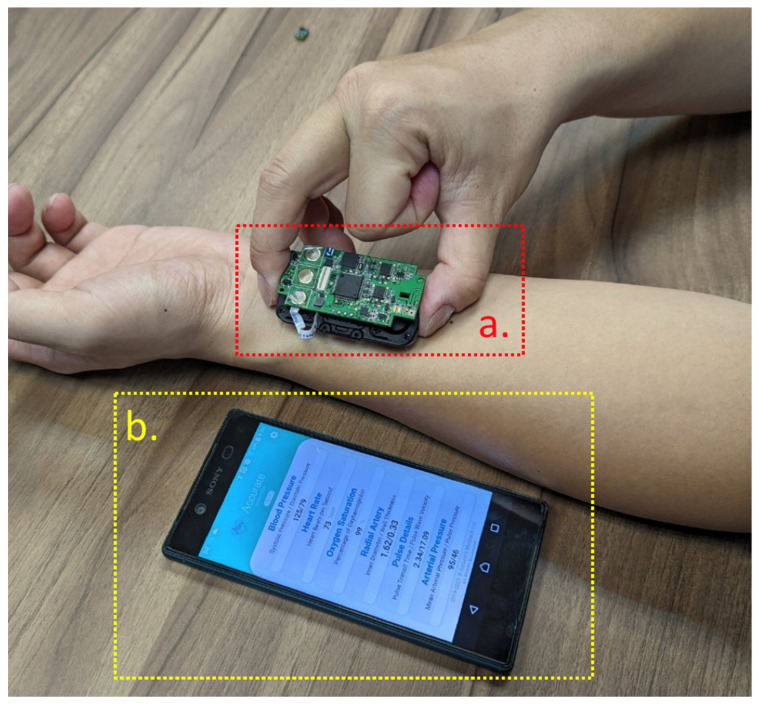
A patient that has been measured. (**a**) The cuffless BP monitor. (**b**) A smartphone displaying the patient’s BP data and recording.

**Figure 3 micromachines-13-01327-f003:**
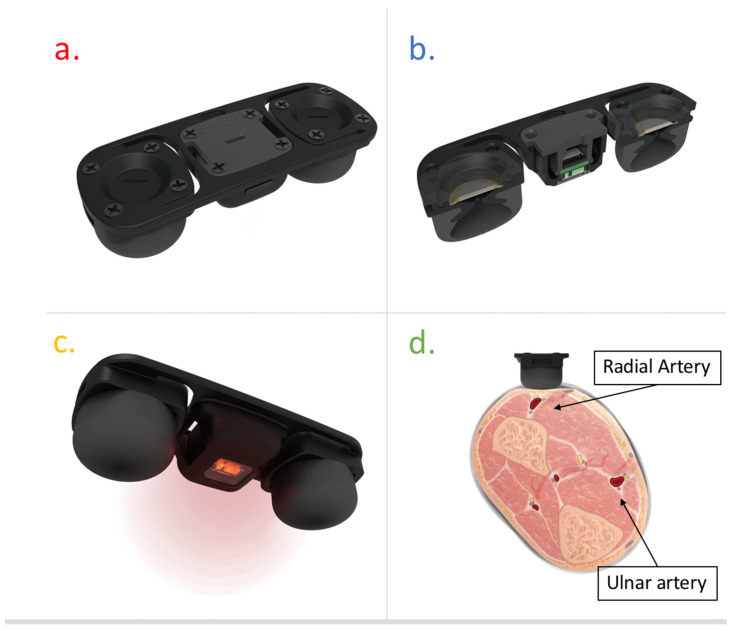
The cuffless BP monitor uses a multi-sensing module that integrates piezoelectric and optical sensors. (**a**) Top view. (**b**) Side view. (**c**) Bottom view. (**d**) The sensor module must be placed parallel above the radial artery.

**Figure 4 micromachines-13-01327-f004:**
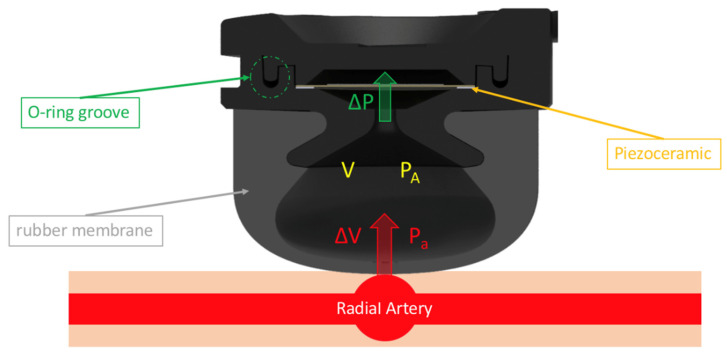
Schematic diagram of the structure of the piezoelectric sensor and the pulse wave transmitted by the radial artery.

**Figure 5 micromachines-13-01327-f005:**
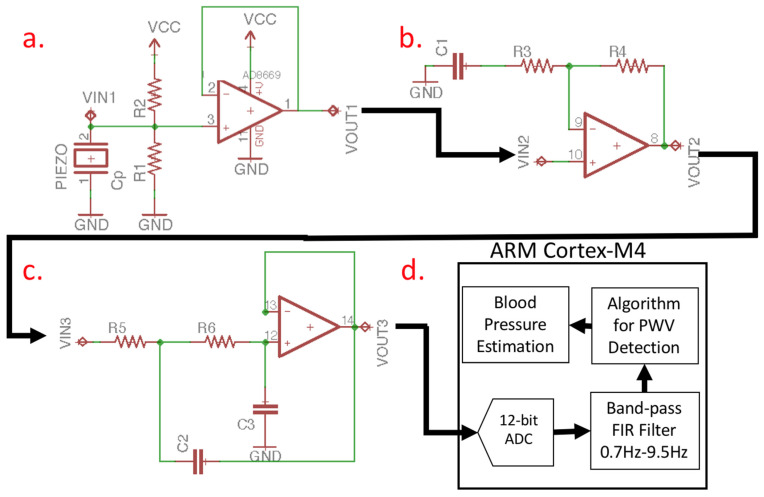
Topology of the AFE on the motherboard. (**a**) The voltage follower (VF). (**b**) The signal amplifier. (**c**) The second-order Sallen–Key low-pass filter (LPF). (**d**) The ARM Cortex-M4.

**Figure 6 micromachines-13-01327-f006:**
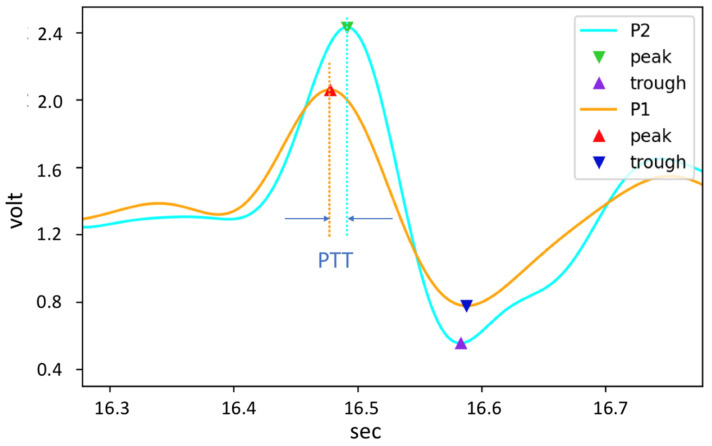
Characteristic points of the pulse wave.

**Figure 7 micromachines-13-01327-f007:**
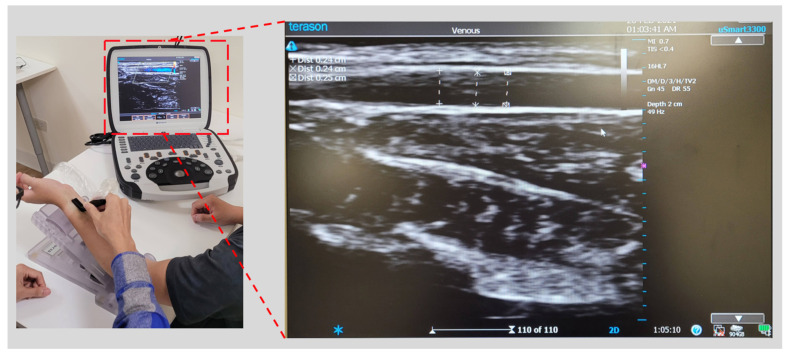
Ultrasound image acquisition of radial artery.

**Figure 8 micromachines-13-01327-f008:**
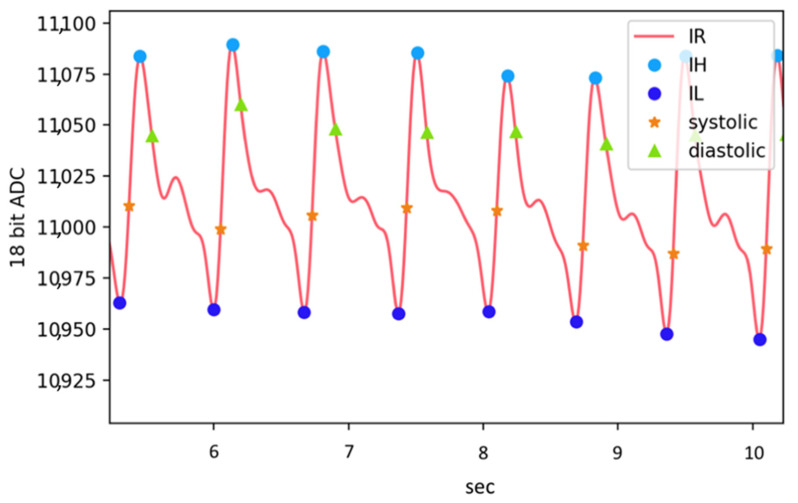
Volume change in the artery.

**Figure 9 micromachines-13-01327-f009:**
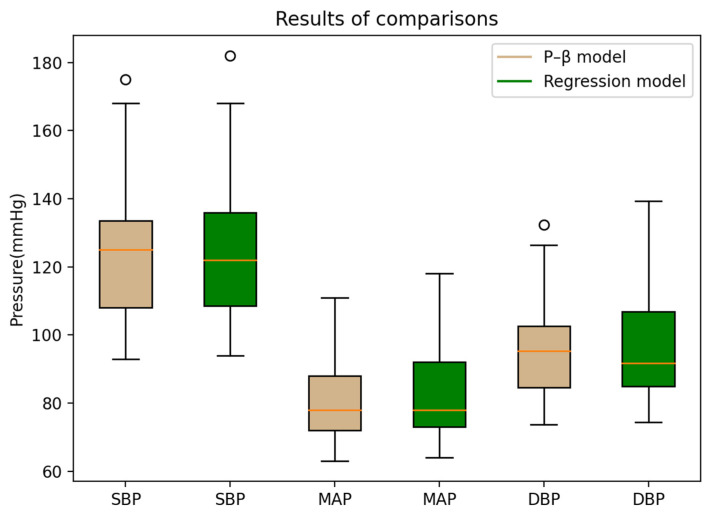
Distribution of BP statistics of the same participants.

**Figure 10 micromachines-13-01327-f010:**
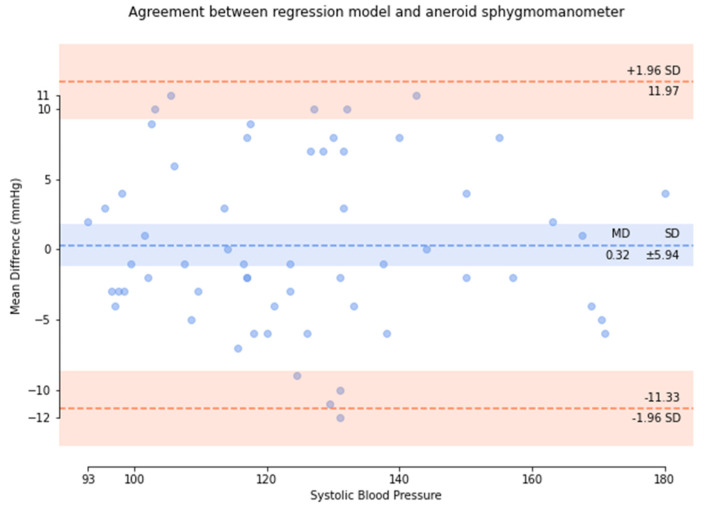
Results of the regression model method for the comparison of the Bland–Altman analysis of SBP.

**Figure 11 micromachines-13-01327-f011:**
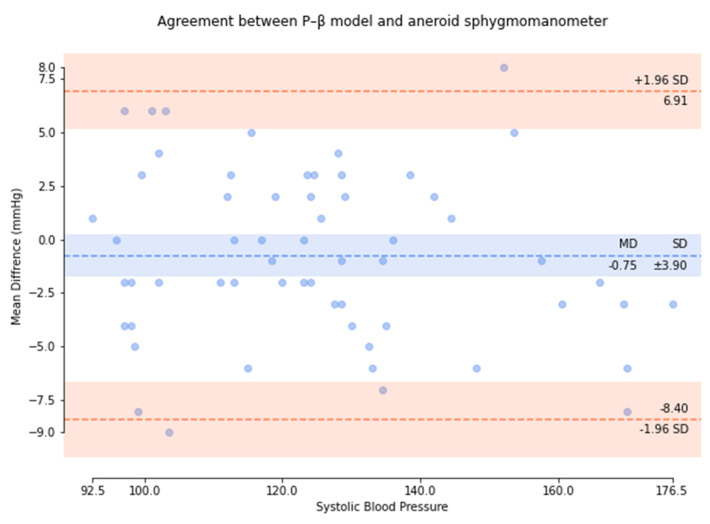
P–β model method for the comparison of the Bland–Altman analysis of SBP.

**Figure 12 micromachines-13-01327-f012:**
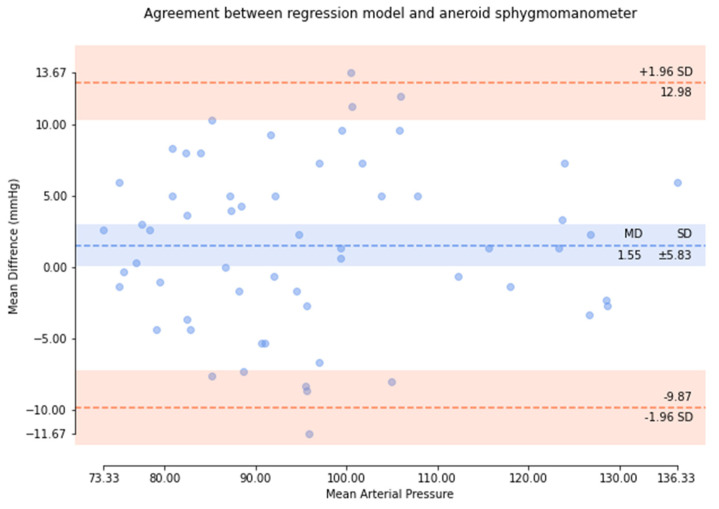
Results of the regression model method for the comparison of the Bland–Altman analysis of MAP.

**Figure 13 micromachines-13-01327-f013:**
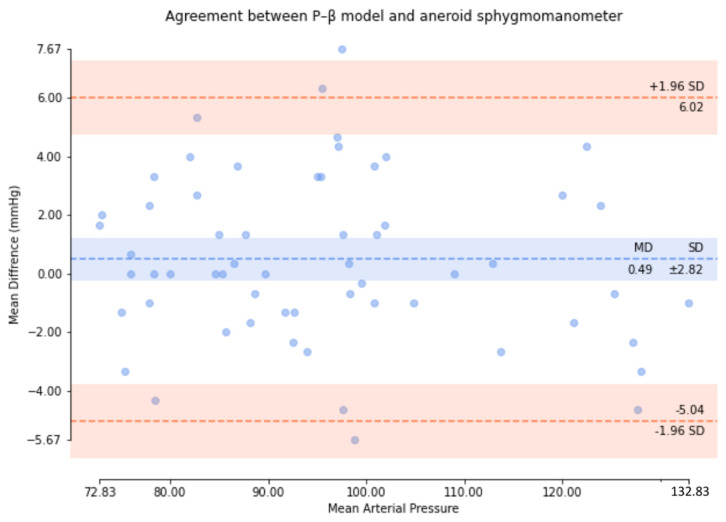
P–β model method for the comparison of the Bland–Altman analysis of MAP.

**Figure 14 micromachines-13-01327-f014:**
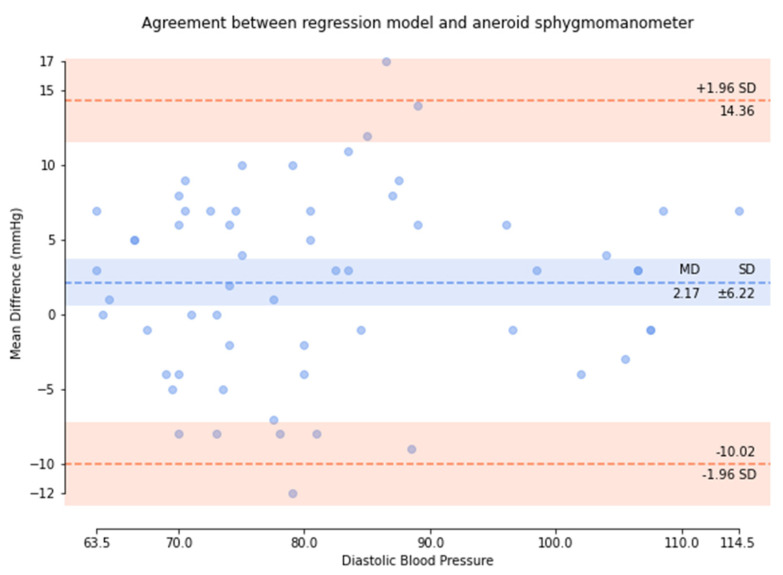
Results of the regression model method for the comparison of the Bland–Altman analysis of DBP.

**Figure 15 micromachines-13-01327-f015:**
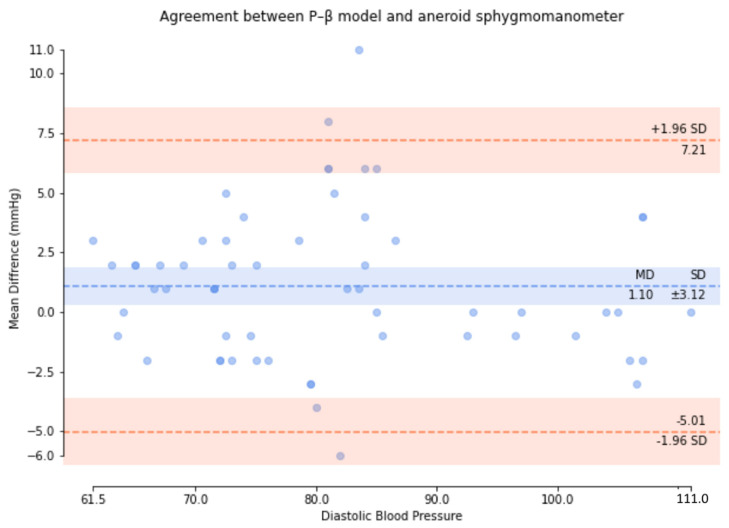
P–β model method for the comparison of the Bland–Altman analysis of DBP.

**Figure 16 micromachines-13-01327-f016:**
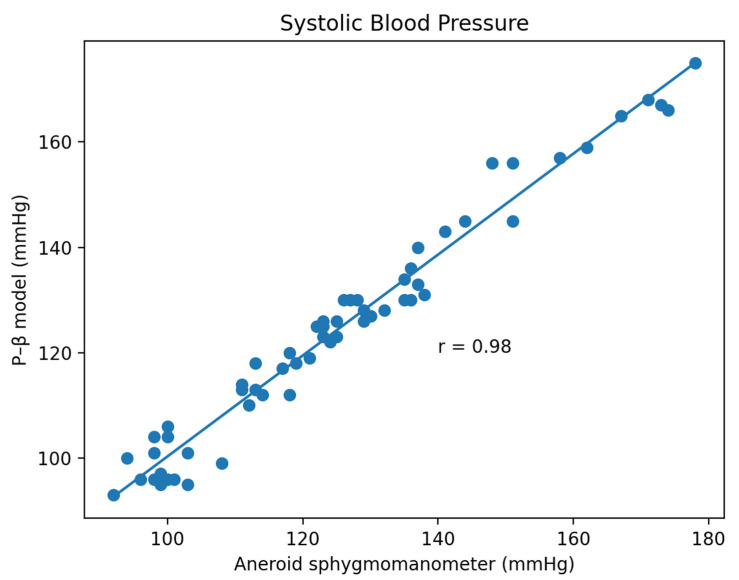
Pearson Correlation of SBP.

**Figure 17 micromachines-13-01327-f017:**
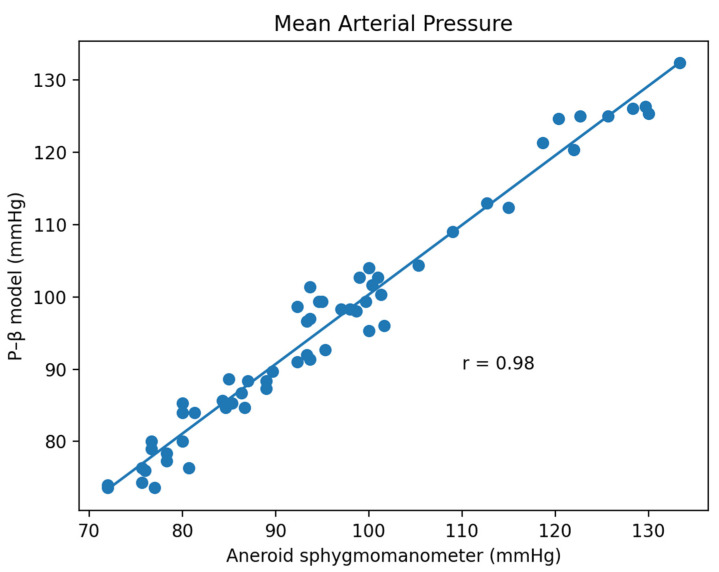
Pearson Correlation of MAP.

**Figure 18 micromachines-13-01327-f018:**
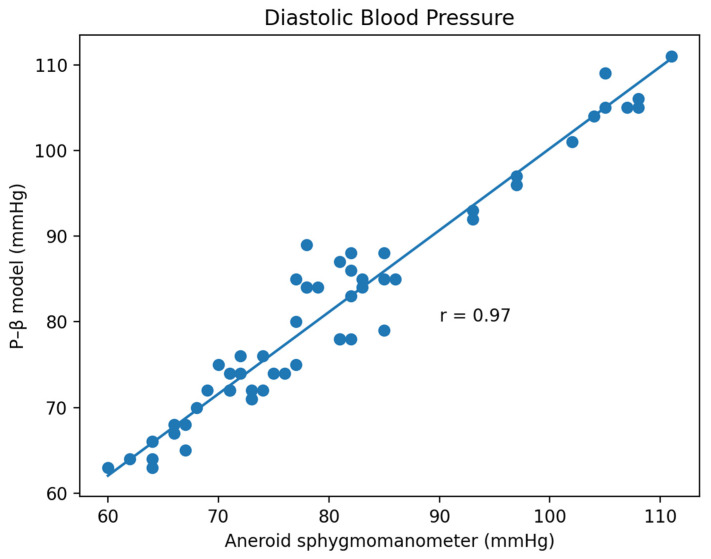
Pearson Correlation of DBP.

**Table 1 micromachines-13-01327-t001:** Validation study results of P–β model method.

	Pass Requirements	Achieved		
		*SBP*	*DBP*	*MAP*
**Criterion 1**				
Means (mmHg)	≤5	0.75	0.49	1.10
SD (mmHg)	≤8	3.90	2.82	3.12
**Results**		**pass**	**pass**	**pass**

**Criterion 2**				
SD (mmHg)	≤6.89/6.92/6.86	3.71	2.67	2.91
**Results**		**pass**	**pass**	**pass**

**Results**		**pass**		

## Data Availability

Not applicable.

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
