# Peer review of "An Arterial Compliance Sensor for Cuffless Blood Pressure Estimation Based on Piezoelectric and Optical Signals"

_micromachines, 2022, doi:10.3390/mi13081327_

Round 1

Reviewer 1 Report

Dear Authors,

There is a significant research in wearables particularly to your topic and based on piezoelectric and optical signals i have the following few concerns:

1. How can you justify your research as compared to https://www.nature.com/articles/s41598-021-03612-1 and https://cjasn.asnjournals.org/content/15/10/1531. I am not asking about the signals and the method but the results and comparative advantages.

2. How your numerical model is in accordance with experimental data (ALL DATA and CASES)?

3.  Is there any other novel factor than Piezo/Opto?

4. How do it pass all the standards ANSI etc?

5. Pl cite the Eq numbers in the text and define all the variables in text.

Reviewer 3 Report

This research developed the well-designed and usable compact BP sensor integrated piezoelectric and optical sensor. 

I accept the article for publication in Micromachines.

Round 2

Reviewer 2 Report

The article can be accepted in its present form.